# Interpretable Function Approximation with Gaussian Processes in Value-Based Model-Free Reinforcement Learning

Matthijs van der Lende[1], Matthia Sabatelli[1], and Juan Cardenas-Cartagena[*1,2]

[1]Bernoulli Institute, University of Groningen, Groningen, The Netherlands.
[2]Wisenet Center, University of Agder, Grimstad, Norway.

## Abstract

Estimating value functions in Reinforcement Learning (RL) for continuous spaces is challenging. While traditional function approximators, such as linear models, offer interpretability, they are limited in their representation capabilities. In contrast, Deep Neural Networks (DNN) can model more complex functions but are less interpretable. Gaussian Process (GP) models bridge this gap by offering interpretable uncertainty estimates while modelling complex nonlinear functions. This work introduces a Bayesian nonparametric framework using GPs, including Sparse Variational (SVGP) and Deep GPs (DGP), for off-policy and on-policy learning. Results on popular classic control environments show that SVGPs/DGPs outperform linear models but converge slower than their DNN counterparts. Nevertheless, they do provide valuable insights when it comes to uncertainty estimation and interpretability for RL[1].

## 1 Introduction

Initially, Reinforcement Learning (RL) relied on tabular methods, which involve storing state-action values in a table. The classical Q-learning algorithm [1] is an example of the tabular approach. This method is sufficient for tasks, where the number of states and actions is small. However, tabular methods do not scale well in high-dimensional environments. Indeed, a critical challenge in RL is approximating value functions and policies, especially in environments with continuous state spaces. Sutton [2] introduced a linear regression-based extension of the Temporal Difference learning algorithm–TD($\lambda$).

While simple and interpretable, traditional linear models often fall short in capturing the behaviour patterns of complex environments, particularly when there are nonlinear relationships in the state-action spaces. The sub-field of Deep RL (DRL) seeks to address this problem by employing Deep Neural Networks (DNN) as function approximators. State-of-the-art DRL algorithms are Deep Q-Networks (DQN) [3] and Proximal Policy Optimization (PPO),

[4]. It is well-known that unlike other machine learning families, such as decision trees and support vector regression, DNN models lack interpretability [5] as, unlike a linear model, there is no linear relationship between each weight and one feature. Nevertheless, popular DNN models, like Convolutional Neural Networks, are widely used because they can learn a broad range of functions and extract features from data, as shown in tasks like pixel-based control from Atari images. [3].

A candidate choice for interpretable function approximation is a Gaussian Process (GP), a nonparametric kernel-based model for regression or classification. Unlike linear models, they can model complex nonlinear functions [6, 7], and their nonparametric nature allows them to scale in complexity with the dataset size. Additionally, GPs provide uncertainty quantification out of the box, as the predictive variance is directly derived from the model, offering a way to estimate uncertainty in predictions. Engel et al. [8] employed GPs to approximate the value and action-value function, resulting in a variant of the SARSA algorithm for on-policy RL. This approach was extended to the off-policy case by Chowdhary et al. [9], who proposed a variant of Q-learning using GPs alongside a proof of convergence. Building on these ideas, Grande et al. [10] derived sample complexity results for GPs in RL and introduced Delayed GPQ, a sample-efficient, model-free RL algorithm using GPs, which can obtain an optimal policy in a polynomial number of exploration steps in continuous state spaces. These methods relied on a sparsification method from Csató and Opper [11] to reduce computational complexity. However, they did not make use of more recent inducing point methods initially introduced by Titsias [12], which offer a more effective and scalable sparsification approach that more accurately approximates the GP model on the full dataset [13]. Kameda and Tanaka [14] applied such inducing point methods to reduce the computational complexity in GP Q-learning.

Previous research on GPs in value function approximation mainly explored strategies like upper confidence bound and $\epsilon$-greedy, without considering alternatives like Thompson sampling, which could improve exploration effectiveness. Moreover, extensions to the basic GP model, such as Deep Gaussian Processes (DGPs) or Sparse Variational Gaussian

---

*Corresponding Author: j.d.cardenas.cartagena@rug.nl

[1]Code is available at: github.com/matthjs/BachelorProject

Proceedings of the 6th Northern Lights Deep Learning Conference (NLDL), PMLR 265, 2025.

Processes (SVGPs), have not been considered despite their ability to capture complex relationships through their hierarchical structure. Additionally, prior work, such as that of Kameda and Tanaka [14], required pre-specified inducing points, which limits the algorithm's applicability in environments where selecting inducing points in advance is challenging. In contrast, the proposed algorithm is more general for online tasks, supports any GP model and learns the inducing points dynamically [15], a key improvement over earlier work. This advancement removes the need to pre-define inducing points, making the algorithm applicable to a broader range of environments.

**Contributions:** This work focuses on the advantages and limitations of GPs as function approximators for model-free RL, focusing on GP regression for the action-value function and its performance compared to traditional TD-learning approaches such as DQN. The paper proposes two new GP-based RL algorithms that connect variants of GPs with on/off-policy RL for online control and explores different exploration strategies. The results show that GPs outperform linear models but underperform deep neural networks.

## 2 Theoretical Framework

### 2.1 Reinforcement Learning

RL is a method for learning policies in sequential decision-making problems, often modelled as Markov Decision Process (MDP) [16]. An MDP is a 4-tuple $\mathcal{M} = (\mathcal{S}, \mathcal{A}, \mathrm{p}, \mathrm{r})$, where $\mathcal{S}$ is a set of states, $\mathcal{A}$ is a set of actions, $\mathrm{p} : \mathcal{S} \times \mathcal{A} \times \mathcal{S} \rightarrow [0, 1]$ is a transition function, and $\mathrm{r} : \mathcal{S} \times \mathcal{A} \rightarrow \mathbb{R}$ is a reward function. The behaviour of the RL agent is modelled by its policy, which is a conditional probability distribution over actions given the state:

$$\pi(a|s) = \mathbb{P}(A_t = a | S_t = s) \quad \forall s, a \in \mathcal{S} \times \mathcal{A}, \quad (1)$$

where $S_t \in \mathcal{S}, A_t \in \mathcal{A}$ are Random Variables (RVs) for the state and action at time $t$. The goal is to learn an optimal policy $\pi^*$ that maximizes the expected discounted sum of rewards,

$$\pi^* = \underset{\pi}{\mathrm{argmax}}\, \mathbb{E}_\pi[G_t] = \underset{\pi}{\mathrm{argmax}}\, \mathbb{E}_\pi\left[\sum_{k=0}^{\infty} \gamma^k R_{t+k+1}\right], \quad (2)$$

where $G$ is the return, $R$ is the reward and $\gamma \in (0, 1]$ is a discount factor. The hypothesis in RL is that by interacting with the environment the agent can learn to make (sub-)optimal decisions. Value-based methods achieve it by learning a value function:

$$v_\pi(s) = \mathbb{E}_\pi[G_t | S_t = s], \quad (3)$$

or an action-value function:

$$q_\pi(s, a) = \mathbb{E}_\pi[G_t | S_t = s, A_t = a], \quad (4)$$

and then deriving a policy from it. We want then an approximation $\hat{v}$ or $\hat{q}$ that approaches the optimal value function and generalizes well. In the case of a parametric model, we learn parameters $\mathbf{w} \in \mathbb{R}^d$.

### 2.2 Gaussian Processes

A GP is a collection of RVs $\{f_{\mathrm{GP}}(\mathbf{x}) \mid \mathbf{x} \in \mathcal{X}\}$, any finite number of which has a joint Gaussian distribution [6]. In the context of regression, the index set $\mathcal{X}$ is related to the input of some function $f : \mathcal{X} \rightarrow \mathbb{R}$ that we want to approximate, e.g., the action-value function. It can be interpreted as: At each point $\mathbf{x} \in \mathcal{X}$, the output of the GP regression model is an RV denoted $f_{\mathrm{GP}}(\mathbf{x})$.

A GP, denoted $f_{\mathrm{GP}}(\mathbf{x}) \sim \mathcal{GP}(m(\mathbf{x}), k(\mathbf{x}, \mathbf{x}'))$, is fully specified by a mean function $m : \mathcal{X} \rightarrow \mathbb{R}$ and covariance or kernel function $k : \mathcal{X} \times \mathcal{X} \rightarrow \mathbb{R}$ which are defined as:

$$\begin{aligned} m(\mathbf{x}) &= \mathbb{E}[f_{\mathrm{GP}}(\mathbf{x})], \\ k(\mathbf{x}, \mathbf{x}') &= \mathbb{E}[(f_{\mathrm{GP}}(\mathbf{x}) - m(\mathbf{x}))(f_{\mathrm{GP}}(\mathbf{x}') - m(\mathbf{x}'))]. \end{aligned} \quad (5)$$

A GP offers a Bayesian approach to nonparametric regression. Without any data, the kernel function represents our prior belief about the function we are trying to model, by abuse of notation denoted $p(f)$, as it encodes similarity between data points, with closer points having higher covariance. A GP can then be conditioned on a dataset, $\mathcal{D}$, to get a posterior GP $f_{\mathrm{GP}}(\mathbf{x}) \sim \mathcal{GP}(m_{\mathrm{post}}(\mathbf{x}), k_{\mathrm{post}}(\mathbf{x}, \mathbf{x}'))$, which is our posterior belief about the function, $p(f|\mathcal{D})$. The Appendix B provides a comprehensive discussion of GPs, and their extensions: Sparse Variational (SVGP) and Deep GPs (DGP).

### 2.3 Value-based RL using Gaussian Processes

Using GPs for value-based RL is done by capturing a distribution over possible $q$-functions, $q_\pi \sim \mathcal{GP}(m(z), k(z, z'))$, where the input domain is $z \in \mathcal{S} \times \mathcal{A}$ [9]. The core idea behind the GP-Q algorithm (see Algorithm 1) is to perform a type of Bayesian optimization on the TD error by setting the target values as described in Algorithm B.1 to the TD(0) target. Using different acquisition functions, we now have more options in designing the behavioural policy other than just using $\epsilon$-greedy. [9] for example used a variant of GP-Q using UCB.

The GP-Q algorithm is off-policy and uses

$$R_{t+1} + \gamma \max_{a \in \mathcal{A}} \bar{q}(S_{t+1}, a) \quad (6)$$

as the TD target, where $\bar{q}$ is the mean output of the GP [9]. An on-policy variant, GP-SARSA, can be

created by setting the target to

$$R_{t+1} + \gamma \bar{q}(S_{t+1}, A_{t+1}). \qquad (7)$$

## 2.4 Interpreting The GP Posterior Predictive Distribution for Action-Value Functions

GPs are known to provide well-calibrated predictive distributions which effectively capture epistemic uncertainty [7]. This means that the predictive variance reflects the confidence in the output estimate with higher variance indicating greater uncertainty in the prediction and lower confidence. For samples outside the train set the uncertainty increases accordingly. If a GP is used to predict action-values, then for each state action pair we have a mean action-value $\bar{q}(s,a)$ and variance $\sigma^2(s,a)$. Small $\sigma^2(s,a)$ indicates that the GP-based RL agent is highly confident about the value of action $a$ in state $s$. Conversely high $\sigma^2(s,a)$ indicates high uncertainty in the predictions, corresponding to under-exploration of that particular part of the state space [17].

This has two key applications. First, it can reduce the number of interactions with the environment required to learn policies, improving sampling efficiency. Second, safe RL utilizes uncertainty quantification to adhere to safety constraints during the learning process [18], particularly vital in robotics applications where an aggressive exploration phase could lead to physical harm of the system.

---

**Algorithm 1** Online GP-Q for estimating $p(q_*|\mathcal{D})$
___
1: Collect initial dataset $\mathcal{D}_0 = (z_i, y_i)_{i=1,\dots,n_0}$ with $z_i \in \mathcal{S} \times \mathcal{A},\ y_i \in \mathbb{R}$
2: Initialize GP by computing $p(q|\mathcal{D}_0)$
3: **for** each time step $t$ **do**
4:     Choose $A_t$ from $S_t$ using behavioral policy (e.g., $\epsilon$-greedy)
5:     Take action $A_t$, observe $R_{t+1}, S_{t+1}$
6:     let $Z_t = (S_t, A_t)^\top$ and $y_t = R_{t+1} + \gamma \max_{a'} \bar{q}(S_{t+1}, a')$, where $\bar{q}$ is the posterior mean function
7:     Add state-action pair to dataset, $\mathcal{D}_{t+1} = \mathcal{D}_t \cup \{(Z_t, y_t)\}$
8:     **if** $|\mathcal{D}_{t+1}| >$ Budget **then**
9:         Delete some $z_i \in \mathcal{D}_{t+1}$
10:     **end if**
11:     Update model by computing $p(q|\mathcal{D}_{t+1})$
12: **end for**

---

# 3 Methodology

## 3.1 Training Environments

Since we are comparing value-based algorithms that involve taking an (arg-)max over the action space, we are constrained to environments with discrete action spaces, however, the state spaces are continuous as we consider the realm of approximate RL. For our experiments, we rely on two popular classic control environments vastly used within the literature: `CartPole` and `Lunar Lander` from the `Gymnasium` library [19]. Note that when it comes to the state representation of the environments we normalize the state vector, by relying on standardization

$$\mathbf{x}_{\text{norm}} = \frac{\mathbf{x} - \bar{\mathbf{x}}}{\sqrt{\sigma_{\mathbf{x}}^2 + \eta}}, \qquad (8)$$

where $\bar{\mathbf{x}}$ and $\sigma_{\mathbf{x}}^2$ are the mean and variance of $\mathbf{x}$ and $\eta$ is a small constant. One exception is that for the `Lunar Lander` environment, we only normalized the first 6 dimensions as some state components are booleans.

## 3.2 Algorithm Design

The GP-Q [9] and GP-SARSA [8] algorithms were adopted for the off-policy and on-policy cases respectively with the following modifications: The updates were done in batches, instead of per timestep. Meaning, that for a batch size $B$, we wait $B$ timesteps before updating the GP. This was done to reduce computational complexity and improve sampling efficiency. Furthermore, this aimed to prevent the inaccurate estimate of the $q$-function from changing too much per timestep. All the modifications made to the base GP-Q/GP-SARSA algorithm are summarized in algorithm 2.

Initially, $\mathcal{D}_0 = \emptyset$. Updating the GP, i.e., computing $p(q|\mathcal{D}_{n+1})$, should be interpreted as performing type-II Maximum Likelihood Estimation (MLE) of the hyperparameters and observation noise variance $\theta' = [\theta, \sigma_y^2]^\top$, and then computing the predictive posterior distribution where the test inputs are the state-action pairs of the current batch. In the case of SVGP/DPGs, we also have the Evidence Lower Bound (ELBO) with variational parameters $\mathbf{Z}, \mathbf{m}, \mathbf{S}$ as hyperparameters. We utilized the Adam optimizer [20] to estimate $\theta'_{\text{opt}}$ by minimizing the negative Marginal Log Likelihood (MLL) (18) or the negative ELBO (23). It was implicitly assumed that any hyperparameter values after optimization carry over to the next GP update. As explained in Appendix B.4, SVGPs allow for mini-batching in ELBO computation, which helps manage memory usage and enables handling significantly larger datasets. Two viable approaches to mini-batching in terms of time complexity are: using mini-batches over the entire dataset while assuming a relatively small dataset and hoping the variational parameters retain information from previously discarded data points; or maintaining a larger dataset and optimizing over a random subset of mini-batches each iteration. In the case of the former, the SVGP

**Algorithm 2** Adjusted GP-Q/GP-SARSA

1: **Initialize:**
2:    Collect initial dataset $\mathcal{D}_0 = (z_i, y_i)_{i=1,\ldots,n_0}$ with $z_i \in \mathcal{S} \times \mathcal{A}$, $y_i \in \mathbb{R}$
3:    Initialize GP with $p(q|\mathcal{D}_0)$, noise variance $\sigma_y^2$, and hyperparameters $\theta$
4:    Set $\theta'_n = [\theta, \sigma_y^2]^\top$ and batch counter $b = 0$
5: **for** each time step $t$ **do**
6:    Choose $A_t$ from $S_t$ using a behavioral policy (e.g., $\epsilon$-greedy, UCB, Thompson sampling)
7:    Take $A_t$, observe $R_{t+1}$ and $S_{t+1}$
8:    $b = b + 1$
9:    **if** Off-Policy **then**
10:        $y_t = R_{t+1} + \gamma \max_{a'} \bar{q}(S_{t+1}, a')$
11:    **else** {On-Policy}
12:        $A_{t+1} = \pi(S_{t+1})$
13:        $y_t = R_{t+1} + \gamma \bar{q}(S_{t+1}, A_{t+1})$
14:    **end if**
15:    **Update dataset:**
16:        Form state-action pair $Z_t = [S_t, A_t]^\top$
17:        $\mathcal{D}_{t+1} = \mathcal{D}_t \cup \{(Z_t, y_t)\}$
18:    **if** $|\mathcal{D}_{t+1}| >$ Budget **then**
19:        Delete oldest $(z_i, y_i)$ from $\mathcal{D}_{t+1}$
20:    **end if**
21:    **if** $b ==$ batch size **then**
22:        **Update GP model,** $p(q|\mathcal{D}_{n+1})$**:**
23:            Minimize negative MLL in (18) or negative ELBO in (23) w.r.t. $\mathcal{D}_{n+1}$
24:            Update hyperparameters to $\theta'_{n+1}$
25:            Reset batch counter: $b = 0$
26:    **end if**
27: **end for**

objective becomes:

$$\mathcal{L}_{\text{SVGP}} = \left[ \frac{N}{B} \sum_{b=1}^{B} \frac{1}{|\mathcal{B}_b|} \sum_{n \in \mathcal{B}_b} \mathbb{E}_{\hat{p}(f_n)}[\log p(y_n|f_n)] \right] \\ - D_{\mathbb{KL}}(\hat{p}(\mathbf{u}) \parallel p(\mathbf{u}|\mathbf{Z})), \quad (9)$$

where $\mathcal{B}_b$ is the $b$'th batch, and $B$ is the number of batches.

Regarding the removal and addition of data points; data points were added unconditionally in a first-in-first-out (FIFO) manner. If the maximum dataset size (budget) is exceeded then the oldest data points are removed. It is worth mentioning that for SVG-Ps/DPGs it is feasible to have a large dataset size of over $100,000$ data points while for exact GPs there are actual VRAM limitations for dataset sizes greater than 1000. The following behavioural policies were considered: $\epsilon$-greedy, UCB and Thompson sampling. UCB was implemented by taking (19) with $\mathbf{x} \in \mathcal{S} \times \mathcal{A}$. Thompson sampling was extended from the Bandit case by selecting the highest value action from the sample $q$-function for a fixed state $s \in \mathcal{S}$. This was done by considering the points $\{(s, a) \mid a \in \mathcal{A}\}$ sampling from the GP latent RVs, $\{f_{\text{GP}}(s, a) \mid a \in \mathcal{A}\}$, and selecting the action $a$ for

for which the sampled $q$-value is highest.

## 3.3    Experimental Setup

We compared GP-Q with DQN using a Multi-Layer Perceptron (MLP) and a linear model, as well as a random policy during training and evaluation. In `Lunar Lander`, we also included GP-SARSA. Since GP-Q and GP-SARSA were expected to perform similarly, we focused mainly on GP-Q. All algorithms were trained for 1000 episodes on `CartPole` and 3000 episodes on `Lunar Lander` with evaluation over 30 episodes. The episode numbers were chosen such that all algorithms would converge.

This comparison involved examining the reward curve/return graph for policy convergence and performance, including the maximum return obtained, and measuring computational complexity and energy usage during training and evaluation[2]. Time complexity was measured by considering execution time during training and evaluation. Space complexity was measured by examining VRAM usage during training. Energy usage was measured in kilojoules (kJ) during training and evaluation. Additionally, we measured the average negative ELBO for each environment over the number of GP updates.

At evaluation time, we also compared the means of returns. A two-sample one-tailed Wilcoxon rank-sum test was performed to compare the average return between a specific RL agent and a random policy after training. The null hypothesis was $H_0 : \mathbb{E}_{\pi_{\text{agent}}}[G] = \mathbb{E}_{\pi_{\text{random}}}[G]$ and the alternative was $H_A : \mathbb{E}_{\pi_{\text{agent}}}[G] > \mathbb{E}_{\pi_{\text{random}}}[G]$, where $G$ is the undiscounted sum of rewards over an episode.

We also measured the stability of the GPQ with a DGP by examining how many times out of $C_r$ runs of at least $C_e$ episodes the algorithms performed better than random. For `CartPole` $C_r = 10$, $C_e = 200$ and `Lunar Lander` $C_r = 7, C_e = 1000$. If an algorithm is stable, we expect the same hyperparameter configuration and environment to produce consistent results w.r.t. random in most runs.

Finally, we recorded the agent environment interaction after training and described the general behaviour in cases where it was insightful. For details about model architecture, experimental design, and hyperparameter tuning, refer to Appendix D.

## 4    Results

The following showcases the results from the experiment described in Section 3.3. A comparison of exploration methods is found in Appendix C.1.

The reward curves are displayed in Figure 1, reward metrics in Table 1, and resource consump-

---

[2]As GP-Q and GP-SARSA are the same, except for the computation of the TD target, such measurements were omitted from GP-SARSA for brevity.

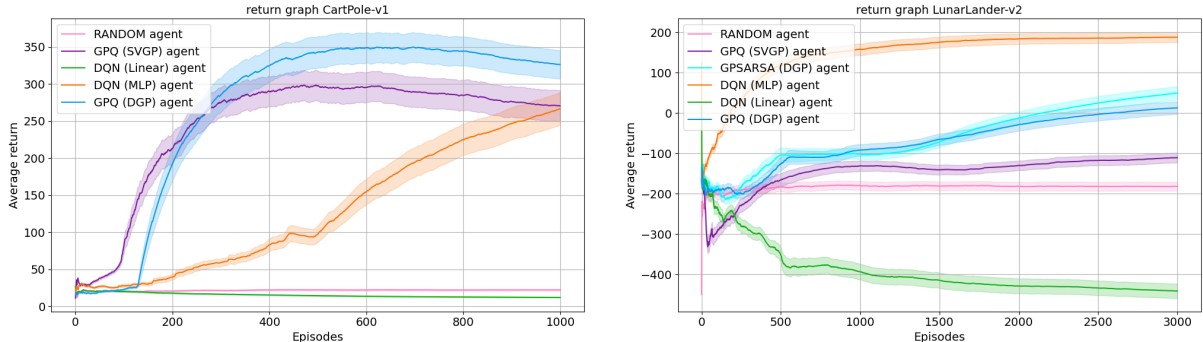

**Figure 1.** Comparison of cumulative average return ($\pm 0.1 \cdot$ std) in CartPole (left) and Lunar Lander (right).

**Table 1.** Performance metrics for `CartPole` and `Lunar Lander`. Values are presented as Train/Evaluation for Average Return and Max Return. Sig. BTR: significantly better (avg. return) than random (Wilcoxon test, $p < 0.05$)

| Algorithm | Avg. Return | Max Return | Sig. BTR |
|---|:---:|:---:|:---:|
| **CartPole (1000 episodes)** | | | |
| DQN (MLP) | $266.43 \pm 219.66$ / $451.53 \pm 147.89$ | 500 / 500 | True |
| **GP-Q (DGP)** | $326.34 \pm 191.95$ / $222.9 \pm 157.24$ | 500 / 500 | **True** |
| **GP-Q (SVGP)** | $270.48 \pm 208.33$ / $276.60 \pm 209.82$ | 500 / 500 | **True** |
| RANDOM | $22.16 \pm 11.83$ / $22.23 \pm 11.27$ | 96 / 52 | NA |
| DQN (Linear) | $12.01 \pm 5.19$ / $9.33 \pm 0.99$ | 55 / 12 | False |
| **Lunar Lander (3000 episodes)** | | | |
| DQN (MLP) | $188.17 \pm 126.85$ / $177.58 \pm 120.07$ | 323.55 / 312.30 | True |
| **GP-Q (DGP)** | $13.09 \pm 156.93$ / $104.06 \pm 128.34$ | 316.62 / 271.29 | **True** |
| **GP-SARSA (DGP)** | $49.69 \pm 164.58$ / $-56.73 \pm 188.40$ | 315.95 / 273.29 | **True** |
| **GP-Q (SVGP)** | $-110.67 \pm 121.80$ / $-99.61 \pm 85.71$ | 269.63 / 142.78 | **True** |
| RANDOM | $-181.95 \pm 110.02$ / $-191.07 \pm 104.94$ | 80.22 / $-71.85$ | NA |
| DQN (Linear) | $-441.17 \pm 178.65$ / $-539.79 \pm 156.03$ | 33.8 / $-186.96$ | False |

tion in Table 2. In the case of `CartPole`, GPQ (SVGP) stopped learning around episode 150, GPQ (DGP) around episode 170, and DQN (MLP) around episode 450. While for `Lunar Lander`, DQN (MLP) stopped learning around episode 600. For GPQ (DGP) and GP-SARSA (DGP), the reward-stopping condition was reached around episode 1500. GPQ (SVGP) did not reach the stopping condition and converged to a sub-optimal policy. DQN (Linear) did not learn any meaningful policy.

The general behaviour of GPQ and DQN (MLP) was similar in `CartPole`; both balanced the pole for the duration of the episode most of the time, with DQN achieving this more often. DQN (Linear) failed to learn any meaningful policy for balancing the pole, resulting in the pole losing its balance immediately. Out of 10 runs, the success rates were 100% for DQN (MLP), 40% for GP-Q (DGP) and 0% for DQN (Linear). And for `Lunar Lander`, GP-Q (DGP) also landed the module reasonably well. GP-Q (DGP) kept the module upright and typically landed it successfully, though it applied more upward force, resulting in a more gradual and slow landing, compared to DQN, which was almost free-falled

initially. GP-Q (SVGP) did not control the module as well as GP-Q (DGP) and as a result, was not as successful in landing the module at the landing spot. Out of 7 runs, the success rate was 100% for DQN (MLP), 100% for GP-Q (DGP) and 0% for DQN (Linear).

It takes around 1000 more episodes for GP-Q/GP-SARSA to reach a close-to-optimal policy in Lunar Lander compared to CartPole, which may be due to the larger state space. This raises questions about how well GP-Q and GP-SARSA converge as the state and action spaces grow larger. However, benchmarks from [15] show that DGPs can handle high-dimensional data, as demonstrated by their application to a 784-dimensional image dataset.

### 4.1 Interpreting GP predictions in Lunar Lander

In the case of Lunar Lander, we visualized the GP posterior distribution by fixing a state $s \in \mathcal{S}$ and plotting a Gaussian with mean $\bar{q}(s, a)$ and variance $\sigma^2(s, a)$ for each $a \in \mathcal{A}$. Figure 2 illustrates these Gaussians, showing that as training progresses, GP-

**Table 2.** Time, memory, and energy usage for `CartPole` and `Lunar Lander` tasks. Values are presented as Train/Evaluation where applicable.

| Algorithm | CartPole (1000 episodes) | | | Lunar Lander (3000 episodes) | | |
|---|---|---|---|---|---|---|
| | Time (s) | Memory (GB) | Energy (kJ) | Time (s) | Memory (GB) | Energy (kJ) |
| **GP-Q (DGP)** | 5995.15 / 121.11 | 1.80 | 252.07 / 4.29 | 27349.92 / 135.99 | 2.35 | 2134.39 / 4.87 |
| **GP-Q (SVGP)** | 2116.31 / 40.02 | 0.566 | 16.11 / 1.57 | 9568.53 / 39.23 | 0.936 | 645.86 / 1.16 |
| DQN (MLP) | 64.01 / 5.48 | 0.414 | 2.15 / 0.16 | 349.91 / 4.22 | 0.678 | 7.92 / 0.093 |
| DQN (Linear) | 12.01 / 0.29 | 0.414 | 0.49 / 0.0087 | 124.44 / 1.95 | 0.02 | 2.79 / 0.042 |

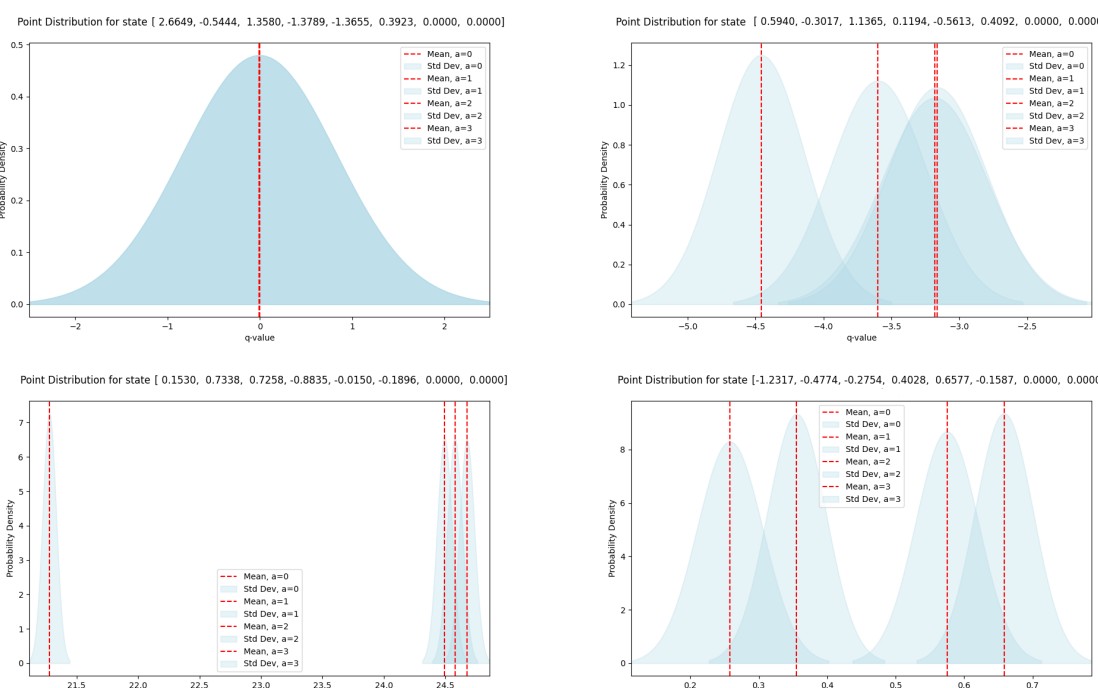

**Figure 2.** Predictive action distribution for a given state for each action $a \in \mathcal{A}$. The top two figures are taken from a GPQ (DGP) agent before training on Lunar Lander, and the bottom two are after training. As training continues, the action distributions shift and the standard deviations around the mean decrease.

Q and GP-SARSA agents using Thompson sampling transition from an exploratory phase, marked by high uncertainty in $q$-value estimates, to an exploitative phase characterized by near-deterministic decision making. Such behaviour indicates that uncertainty plays a key role in the RL agent's learning process as it interacts with the environment.

## 5 Conclusions

The work aimed to compare a GP-based RL algorithm with linear and neural network function approximators, specifically against DQN, using linear and MLP models. The goal was to assess the strengths and limitations of GPs in RL.

Findings from simulations align with the hypothesis: GP-based algorithms (GP-Q/GP-SARSA), particularly using SVGPs and DGPs, outperform linear function approximation in CartPole and Lunar Lander. However, they do not match the stability of DQN with an MLP in CartPole or the overall performance in Lunar Lander. Additionally, GP-Q and GP-SARSA are more computationally expensive, even during inference.

Utilizing uncertainty quantification, GP-based agents via Thompson sampling automatically balance exploration and exploitation, unlike DQN, which relies on random action selection.

These results underscore the potential of GPs, particularly DGPs, as function approximators in RL tasks requiring uncertainty quantification and interpretability, such as safe RL, where understanding the confidence in predictions can mitigate risks and ensure robust decision-making.

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

# A Future Works

An important next step is extending the proposed algorithm to continuous action spaces. Currently, we are limited to discrete actions due to the need to compute an argmax over the action space. This limitation could be addressed by integrating GPs into an actor-critic framework. According to Lockwood and Si [17], uncertainty quantification in RL has mainly focused on the critic (value functions) because it is directly affected by aleatoric (data) uncertainty and this uncertainty propagates to the actor through updates of the form $\nabla_\theta J = \mathbb{E}[\nabla_\theta \log(\pi_\theta)\delta]$, where $\delta$ depends on the critic.

Regarding DGPs, Jankowiak et al. [21] have proposed an alternative parametric model called deep sigma point process (DSPP). This model retains many properties of DGPs without needing to approximate the predictive posterior distribution. Empirical results suggest that DSPPs achieve better-calibrated predictive distributions and can outperform deep kernel learning and DGPs on univariate and multivariate regression tasks. For this reason, it is an important model to consider for future research in GPs and RL.

Lastly, steps must be taken to reduce the overall time complexity, as there is currently a noticeable difference in training and inference speed compared to DQN. Future research can tackle this from two angles: making the algorithm itself more efficient and developing a faster GP model, particularly for online settings.

# B Detailed Theoretical Framework

## B.1 Gaussian Process Derivation

Given a train set of noisy observations $\mathcal{D} = (\mathbf{x}_i, y_i)_{i=1,\dots,N}$, where $\mathbf{x}_i \in \mathcal{X}$ and $y_i = f(\mathbf{x}_i) + \epsilon_i$ with $\epsilon_i \sim \mathcal{N}(0, \sigma_y^2)$. Suppose we are interested in getting predictions $\mathbf{f}_* = [f_{\mathrm{GP}}(\mathbf{x}_1^*), \dots, f_{\mathrm{GP}}(\mathbf{x}_{N_*}^*)]^\top$ [3] for test inputs $\mathbf{X}_* = (\mathbf{x}_1^*, \dots, \mathbf{x}_{N_*}^*)$. By the definitions of a GP, it follows that the prior joint distribution $p(\mathbf{y}, \mathbf{f}_* | \mathbf{X}, \mathbf{X}_*)$ has the following form:

$$p(\mathbf{y}, \mathbf{f}_* | \mathbf{X}, \mathbf{X}_*) = \mathcal{N}\left( \begin{bmatrix} \mu_X \\ \mu_* \end{bmatrix}, \begin{bmatrix} \mathbf{K}_\sigma & \mathbf{K}_{X,*} \\ \mathbf{K}_{X,*}^\top & \mathbf{K}_{*,*} \end{bmatrix} \right), \tag{10}$$

where $\mu_X = [(m(\mathbf{x}_1), ..., m(\mathbf{x}_N)]^\top$, $\mu_* = [m(\mathbf{x}_1^*), \dots, m(\mathbf{x}_{N_*}^*)]^\top$, $\mathbf{K}_{X,X} = k(\mathbf{X}, \mathbf{X}) \in \mathbb{R}^{N \times N}$, $\mathbf{K}_\sigma = \mathbf{K}_{X,X} + \sigma_y^2 \mathbf{I}$, $\mathbf{K}_{X,*} = k(\mathbf{X},, \mathbf{X}_*) \in \mathbb{R}^{N \times N_*}$, and $\mathbf{K}_{*,*} = k(\mathbf{X}_*, \mathbf{X}_*) \in \mathbb{R}^{N_* \times N_*}$ are matrices of all the covariances between relevant datapoints.

One can then condition on the observations to get the Bayesian predictive distribution for the test

---

[3]$\mathbf{f}_*$ is implicitly assumed to be a realization of the random vector.

---

points. By Gaussian identities, we can get this in closed form:

$$p(\mathbf{f}_* | \mathbf{X}, \mathbf{y}, \mathbf{X}_*) = \mathcal{N}(\mu_{*|\mathcal{D}, \mathbf{X}_*}, \mathbf{\Sigma}_{*|\mathcal{D}, \mathbf{X}_*}), \tag{11}$$

where

$$\begin{aligned} \mu_{*|\mathcal{D}, \mathbf{X}_*} &= \mu_* + \mathbf{K}_{X,*}^\top \mathbf{K}_\sigma^{-1} (\mathbf{y} - \mu_X), \\ \mathbf{\Sigma}_{*|\mathcal{D}, \mathbf{X}_*} &= \mathbf{K}_{*,*} - \mathbf{K}_{X,*}^\top \mathbf{K}_\sigma^{-1} \mathbf{K}_{X,*}. \end{aligned} \tag{12}$$

Assuming a zero mean function, this reduces for a single test point to:

$$\begin{aligned} \mathbb{E}[f_{\mathrm{GP}}(\mathbf{x}_*)] &= \mu(\mathbf{x}_*) = \mathbf{k}_*^\top \underbrace{\mathbf{K}_\sigma^{-1} \mathbf{y}}_{\alpha} = \textstyle\sum_{i=1}^N \alpha_i k(\mathbf{x}_i, \mathbf{x}_*), \\ \mathbb{V}[f_{\mathrm{GP}}(\mathbf{x}_*)] &= \sigma^2(\mathbf{x}_*) = k(\mathbf{x}_*, \mathbf{x}_*) - \mathbf{k}_*^\top \mathbf{K}_\sigma^{-1} \mathbf{k}_*, \end{aligned} \tag{13}$$

where $\mathbf{k}_* = [k(\mathbf{x}_*, \mathbf{x}_1), \dots, k(\mathbf{x}_*, \mathbf{x}_N)]^\top$.

To get the posterior mean and covariance function we consider (12) and (13) over an infinite number of potential test points:

$$\begin{aligned} m_{\mathrm{post}}(\mathbf{x}_*) &= m(\mathbf{x}_*) + \mathbf{k}_*^\top \mathbf{K}_\sigma^{-1} (\mathbf{y} - \mu_X) \\ k_{\mathrm{post}}(\mathbf{x}_*, \mathbf{x}_*') &= k(\mathbf{x}_*, \mathbf{x}_*') - \mathbf{k}_*^\top \mathbf{K}_\sigma^{-1} \mathbf{k}_{*'}, \end{aligned} \tag{14}$$

where $\mathbf{k}_{*'} = [k(\mathbf{x}_*', \mathbf{x}_1), \dots, k(\mathbf{x}_*', \mathbf{x}_N)]^\top$.

## B.2 Kernel Learning

The generalization properties of GPs rely on the selection of the appropriate kernel [6, 7]. A common kernel is the Matern kernel, given by

$$k_{\mathrm{matern}}(\mathbf{x}, \mathbf{x}'; l, \nu, \sigma_f) = \sigma_f \frac{2^{1-\nu}}{\Gamma(\nu)} \left(\sqrt{2\nu} d\right)^\nu K_\nu\left(\sqrt{2\nu} d\right), \tag{15}$$

with $d = (\mathbf{x} - \mathbf{x}')^\top l^{-2} (\mathbf{x} - \mathbf{x}')$, lengthscale parameter $l$, outputscale parameter $\sigma_f$, smoothness parameter $\nu$, Gamma function $\Gamma$ and a modified Bessel function $K_\nu$. As $\nu \to \infty$, the Matern kernel approaches the Radial Basis Function (RBF) kernel:

$$k_{\mathrm{RBF}}(\mathbf{x}, \mathbf{x}'; l, \sigma_f) = \sigma_f \exp\left( -\frac{\|\mathbf{x} - \mathbf{x}'\|^2}{2l^2} \right), \tag{16}$$

which when used results in smooth infinitely differentiable functions being sampled.

For generalization, it is also important to optimize the GP hyperparameters, such as kernel parameters and noise variance, alongside computing the predictive distribution as seen in (12). This can be done by performing type-II MLE through maximizing the MLL $\log p(\mathbf{y}|\mathbf{X}, \theta)$, for hyperparameters $\theta$:

$$p(\mathbf{y}|\mathbf{X}, \theta) = \int_{\mathbb{R}^N} p(\mathbf{y}|\mathbf{f}, \mathbf{X}) p(\mathbf{f}|\mathbf{X}, \theta) \, d\mathbf{f}. \tag{17}$$

As $\mathbf{y} \sim \mathcal{N}(\mathbf{0}, \mathbf{K}_\sigma)$ this integral can be computed as:

$$\log p(\mathbf{y}|\mathbf{X}, \theta) = -\frac{1}{2}\mathbf{y}^\top \mathbf{K}_\sigma^{-1} \mathbf{y} - \frac{1}{2}\log|\mathbf{K}_\sigma| - \frac{N}{2}\log(2\pi), \tag{18}$$

where the first term is a data fit term, the second term, a model fit term, and the last is a constant. The negative MLL is differentiable with respect to $\theta$, so stochastic gradient descent can be used.

Using black box matrix-matrix multiplication (BBMM) inference [22], which leverages GPU acceleration, the overall time and space complexity of GP regression is $O(N^2)$. See Appendix B.6 for details.

## B.3 Bayesian Optimization and Multi-Armed Bandits

Bayesian optimization (BayesOpt) concerns itself with global optimization of black-box functions $f : \mathcal{X} \to \mathbb{R}$ [7]. Commonly a GP is used as a regressor or surrogate for $f$ based on the data collected so far.

The BayesOpt algorithm as shown in B.1 proceeds as follows: At each iteration $n$, a dataset $\mathcal{D}_n = (\mathbf{x}_i, y_i)_{i=1,\ldots,n}$ is maintained where the target outputs $y_i = f(\mathbf{x}_i) + \epsilon_i$ are assumed to be noisy outputs of the function $f$ we want to optimize. A GP can then be used to estimate $p(f|\mathcal{D})$, a distribution over $f$. An acquisition function $\alpha(\mathbf{x}; \mathcal{D}_n)$ is then used to select a new candidate $\mathbf{x}$ based on its expected utility. Once $y_{n+1} = f(\mathbf{x}_{n+1}) + \epsilon_{n+1}$ has been observed, the GP is updated by computing $p(f|\mathcal{D}_{n+1})$.

In particular, [23] provided sublinear regret bounds using their GP optimization algorithm with $y_n = R_n + \epsilon_n$ as targets, implying that the algorithm's action selection becomes optimal over time. However, this algorithm is constrained to the bandit's problem, which learns a policy in a single-state, discrete-actions environment.

---

**Algorithm B.1** Bayesian Optimization

1: Collect initial dataset $\mathcal{D}_0 = (\mathbf{x}_i, y_i)_{i=1,\ldots,n_0}$ from random queries $\mathbf{x}_i$ or a space-filling design
2: Initialize model (e.g., a GP) by computing $p(f|\mathcal{D}_0)$
3: **for** $n = 1, 2, \ldots$ until convergence **do**
4:     Choose next query point $\mathbf{x}_{n+1} = \operatorname{argmax}_{\mathbf{x} \in \mathcal{X}} \alpha(\mathbf{x}; \mathcal{D}_n)$
5:     $y_{n+1} = f(\mathbf{x}_{n+1}) + \epsilon_n$
6:     $\mathcal{D}_{n+1} = \mathcal{D}_n \cup \{(\mathbf{x}_{n+1}, y_{n+1})\}$
7:     Update model by computing $p(f|\mathcal{D}_{n+1})$
8: **end for**

---

### B.3.1 Acquisition Functions

The acquisition function regulates exploration in the input space, resembling behavioural policies in RL, designed to favour inputs $\mathbf{x}$ with high uncertainty in $f(\mathbf{x})$ while minimizing selections of already explored points. This approach results in more confident estimates for $f(\mathbf{x})$. Various acquisition functions exist [7]:

**Upper confidence bound** (UCB) is an acquisition function defined as:

$$\alpha_n(\mathbf{x}; \mathcal{D}_n) = \mu_n(\mathbf{x}) + \beta_n \sigma_n(\mathbf{x}), \qquad (19)$$

where $\mu_n, \sigma_n$ are the mean and standard deviation outputs as described in (13). $\beta_n$ is an exploration parameter.

Another acquisition function is **Thompson sampling**. In the context of Bandits, this involves sampling an action-value function $\tilde{q}$ from the GP posterior predictive distribution, and greedily selecting an action according to the sample:

$$a_{n+1} = \operatorname*{argmax}_{a \in \mathcal{A}} \tilde{q}(a) \quad \tilde{q}(\cdot) \sim p(q|\mathcal{D}_n). \qquad (20)$$

The intuition behind Thompson sampling is that exploration is encouraged by maximizing $\tilde{q}$ because the sampled function is within the credible interval (standard deviations around the mean) with high values around the areas with high uncertainty. Maximizing $\tilde{q}$ involves selecting actions where there is potentially high uncertainty on $q$. At the same time, actions with a high mean value are also likely to be sampled, which promotes exploitation.

## B.4 Scaling Gaussian Process Inference to Large Datasets

To address the $O(N^2)$ time and space complexity of exact inference, where $N = |\mathcal{D}|$, different approximation approaches can be taken to allow GPs to scale to larger datasets. This is especially of concern in RL, where in theory we have a continually growing dataset. See [7] for a full overview of the available techniques.

### B.4.1 Sparse Variational Gaussian Processes

SVGPs [7, 12, 13, 21, 24, 25] approximate the GP posterior predictive distribution through variational inference. The core idea is to use a set of inducing points $\mathbf{Z} = (\mathbf{z}_1, \ldots, \mathbf{z}_M)$ where $M \ll N$, which serve as a sparse approximation of the full dataset. The associated inducing variables are denoted $\mathbf{u} = [f_{\mathrm{GP}}(\mathbf{z}_1), \ldots, f_{\mathrm{GP}}(\mathbf{z}_M)]^\top$. The variational posterior is defined as:

$$\hat{p}(\mathbf{f}, \mathbf{u}) = p(\mathbf{f}|\mathbf{u}, \mathbf{X}, \mathbf{Z})\hat{p}(\mathbf{u}), \quad \hat{p}(\mathbf{u}) = \mathcal{N}(\mathbf{m}, \mathbf{S}), \qquad (21)$$

where $p(\mathbf{f}|\mathbf{u}, \mathbf{X}, \mathbf{Z})$ is the conditional density of the function values $\mathbf{f} = f_{\mathrm{GP}}(\mathbf{X})$ given train inputs $\mathbf{X}$, inducing points $\mathbf{Z}$ and inducing variables $\mathbf{u}$. $\hat{p}(\mathbf{u})$ is a Gaussian distribution with mean $\mathbf{m}$ and covariance matrix $\mathbf{S}$.

Variational inference aims to **minimize** the Kullback-Leibler (KL) divergence $D_{\mathbb{KL}}(\hat{p} \parallel p)$ between the variational posterior $\hat{p}$ and the true posterior $p$. Using the variational posterior in (21) and

the lower bound on the marginal likelihood,

$$\mathbb{E}_{\hat{p}(\mathbf{f},\mathbf{u})}\left[\log\frac{p(\mathbf{y},\mathbf{f},\mathbf{u})}{\hat{p}(\mathbf{f},\mathbf{u})}\right], \quad (22)$$

we obtain the evidence lower bound (ELBO):

$$\mathcal{L}_{\mathrm{SVGP}} = \sum_{i=1}^{N}\mathbb{E}_{\hat{p}(f_i)}[\log p(y_i|f_i)] - D_{\mathbb{KL}}(\hat{p}(\mathbf{u}) \parallel p(\mathbf{u}|\mathbf{Z})), \quad (23)$$

where $p(y_i|f_i)$ is the Gaussian likelihood for the observations given latent function values. Since the bound is a sum over the data, an unbiased estimator can be obtained using mini-batch subsampling. The variational parameters $\mathbf{Z}$, $\mathbf{m}$, and $\mathbf{S}$, are estimated by maximizing the lower bound $\mathcal{L}_{\mathrm{SVGP}}$. This approach is guaranteed to converge because $\mathcal{L}_{\mathrm{SVGP}}$ is a lower bound to the MLL, i.e., $\log p(\mathbf{y}|\mathbf{X}) \geq \mathcal{L}_{\mathrm{SVGP}}$.

Posterior predictions for test points are now made by marginalizing over the inducing variables $p(\mathbf{f}_*|\mathcal{D},\mathbf{X}_*) \approx \int_{\mathbb{R}^M} p(\mathbf{f}_*|\mathbf{u})\hat{p}(\mathbf{u})\,d\mathbf{u}$ which results in another multivariate Gaussian. The resulting mean and variance predictions for a test point $\mathbf{x}_*$ become:

$$\begin{aligned}
\mu(\mathbf{x}_*) &= m(\mathbf{x}_*) + \alpha(\mathbf{x}_*)^\top(\mathbf{m} - m(\mathbf{Z})), \\
\sigma^2(\mathbf{x}_*) &= k(\mathbf{x}_*,\mathbf{x}_*) - \alpha(\mathbf{x}_*)^\top(\mathbf{K}_{Z,Z} - \mathbf{S})\alpha(\mathbf{x}_*), \\
\alpha(\mathbf{x}_*) &= \mathbf{K}_{Z,Z}^{-1}k(\mathbf{Z},\mathbf{x}_*),
\end{aligned} \quad (24)$$

where $\mathbf{K}_{Z,Z} = k(\mathbf{Z},\mathbf{Z})$ and $k(\mathbf{Z},\mathbf{x}_*) = [k(\mathbf{x}_*,\mathbf{z}_1),\ldots,k(\mathbf{x}_*,\mathbf{z}_M)]^\top$.

The time complexity for SVGPs is $O(NM^2)$, as it can be shown that the likelihood term in (23) can be computed in $O(NM^2)$ time [7]. In terms of storage, we have an $N \times M$ and $M \times M$ covariance matrix, which is in $O(NM + M^2)$.

SVGPs do not overfit with an increasing number of inducing points, and as $M$ increases, the approximation quality of exact inference is recovered. Too few inducing points may make the GP behave as if it was underfitting [13].

## B.5 Deep Gaussian Processes

Another drawback of GPs is the inability of their kernel functions to handle structured data where the similarity between two data points requires hierarchical feature extraction, which occurs in image data and also some vector datasets [25]. DGPs seek to address this issue, while still staying in a Bayesian nonparametric framework.

A DGP is a composition of GPs [7, 26]:

$$\begin{aligned}
\mathcal{DGP}(\mathbf{x}) &= f_L \circ \cdots \circ f_1(\mathbf{x}), \\
f_i(\cdot) &= [f_{\mathrm{GP},i}^{(1)}(\cdot),\ldots,f_{\mathrm{GP},i}^{(H_i)}(\cdot)]^\top, \quad (25) \\
f_{\mathrm{GP},i}^{(j)} &\sim \mathcal{GP}(m_i(\cdot),k_i(\cdot,\cdot)).
\end{aligned}$$

DGPs have a neural network-like structure with $L$ layers, each containing $H$ GPs. Empirical results suggest that DGPs do not overfit as the number

of layers increases, even with limited data, and additional layers generally improve performance on large datasets [15]. [15] also showed that for the same computational budget, increasing the number of layers can be more effective than increasing the number of inducing points in an SVGP.

One can show that a DGP is strictly more general than a GP [7], as a DGP is not just another GP. That said, posterior inference in DGP is quite expensive, as it requires marginalizing over a large number of RVs, corresponding to the hidden function values at each layer. Additionally, the posterior predictive distribution needs to be approximated using Monte Carlo samples, i.e., a finite mixture of Gaussian distributions.

[15] addressed the former by using a variational approach similar to the SVGP method in Section B.4 to allow DGP to scale to larger datasets. This method is called doubly stochastic variational inference, which is the technique used for DGP modelling in this thesis.

The time and space complexity for DGPs using doubly stochastic variational inference is analogous to SVGPs. The ELBO also has a similar form and takes $O(NM^2(D^1 + \ldots + D^L))$ time to compute for $N$ train samples, $M$ inducing points and where $D^i$ is the number of GPs in layer $i$. Similarly, the space complexity is $O((NM + M^2)(D^1 + \ldots + D^L))$.

## B.6 Complexity Analysis of Exact Gaussian Process Inference

When it comes to computational complexity there are two points of interest: Computing the predictive posterior distribution (13) and computing the MLL (18).

To ensure numerical stability, the Cholesky decomposition of $\mathbf{K}_\sigma = \mathbf{L}_\sigma\mathbf{L}_\sigma^\top \in \mathbb{R}^{N \times N}$ is used, which is in $O(N^3)$. After computing $\alpha = \mathbf{K}_\sigma^{-1}\mathbf{y}$, predictions for each test point take $O(N)$ time for the mean and $O(N^2)$ time for the variance. Space complexity is $O(N^2)$ since an $N \times N$ covariance matrix must be stored.

BBMM inference [22], used in the `GPytorch` library, allows for computing the GP MLL (18) and other expensive GP operations using only matrix multiplication, leveraging GPU acceleration. This reduces the time complexity for exact GP inference from $O(N^3)$ to $O(N^2)$. Note that this does not reduce the space complexity.

## B.7 Complexity Analysis of the GP-Q and GP-SARSA Algorithm Variants

The time and space complexity of GP-Q and GP-SARSA mainly depend on the GP model and how the GP is updated with the dataset. As they only

differ in computing the TD target, their complexities are the same. In detail, the Time and Space complexity of GP-Q and GP-SARSA algorithms for different GP models is given below with $N$: dataset size; $M$: inducing points; $D^i$: GPs in layer $i$:

- **Exact GP (BBMM):**
  - Time Complexity: $O(N^2)$
  - Space Complexity: $O(N^2)$

- **SVGP:**
  - Time Complexity: $O(NM^2)$
  - Space Complexity: $O(NM + M^2)$

- **DGP:**
  - Time Complexity: $O(NM^2 \sum D^i)$
  - Space Complexity: $O((NM + M^2) \sum D^i)$

# C   Additional Results

## C.1   $\epsilon$-greedy vs Upper Confidence Bound vs Thompson Sampling

UCB's $\beta$ parameter (19) was set to 1.5, and the $\epsilon$-greedy schedule was based on DQN for a specific environment.

Thompson sampling outperforms UCB and $\epsilon$-greedy in Figure C.1, making it a preferred policy for the GP-Q/GP-SARSA algorithm. The performance gap in UCB may be due to its tendency for over-exploration, as noted by [9]. Additionally, the $\epsilon$-greedy schedule effective for DQN may not suit GP-Q. Thompson sampling's advantage is the lack of exploration parameters to tune.

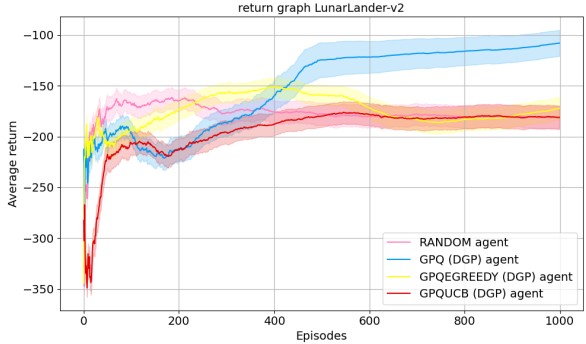

**Figure C.1.** Comparison of cumulative average return ($\pm 0.1 \cdot$std) in Lunar Lander between behavioural policies for the same DGP model.

## C.2   Loss Curve of Experiment

Figure C.2 showcases how the negative ELBO changes per update during training.

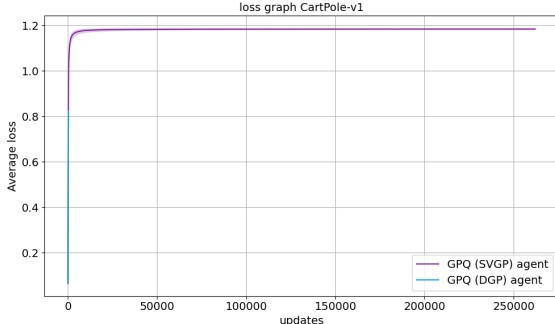

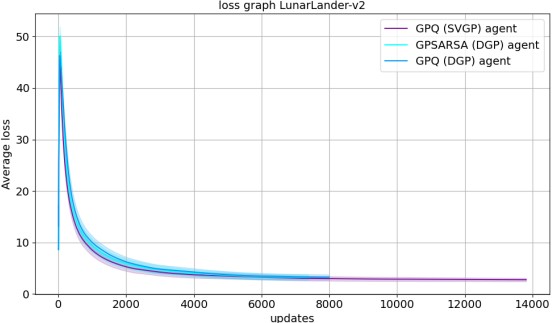

**Figure C.2.** Average loss (negative ELBO) ($\pm 0.1 \cdot$std) in CartPole (top) and Lunar Lander (bottom).

# D   Experimental Design in Detail

## D.1   Model Architectures

The computational backend for all the models is `PyTorch` [27]. For the GP models, we used `GPytorch` [22], a high-performance GPU-accelerated library for GP modelling, in conjunction with `Botorch` [28], a Bayesian optimization library that extends `GPytorch`. An NVIDIA RTX 4090 with 24 GB of VRAM was used for GPU acceleration.

We were also interested in measuring VRAM usage on the GPU, energy consumption, and execution time. For VRAM usage, we used `pynvml`, a Python interface for the NVIDIA Management Library. For energy consumption and execution time, we used the `Zeus` library [29].

### D.1.1   Gaussian Processes and Kernel Selection

In selecting the GP model, we primarily considered DGPs, which reduce to an SVGP when using a single unit.

In Section 2.2, the derivations assumed an observation noise variance parameter $\sigma_y^2$. In the RL setting, it is hard to estimate the exact value of $\sigma_y^2$. To address this, instead of setting $\sigma_y^2$ a priori, we inferred it alongside the hyperparameters by including $\sigma_y^2$ in the optimization process when minimizing the negative MLL in (18) or the negation of the ELBO in (23).

Regarding the choice of the kernel for the states, existing literature on GPs and RL have used the RBF (16) or Matern (15) kernel [9, 14]. This implies that those authors had a prior belief that the action-value function is reasonably smooth. We assumed that this assumption was reasonable. Among the two, the RBF kernel was used.

Regarding [8]'s suggestion to use a separate kernel for the states and action is sensible, but this causes issues with sparse variational methods when learning the inducing points $\mathbf{Z} \in \mathbb{R}^{Md}$ using gradient methods [7]. The actions are discrete, but the problem is that the optimizer that optimizes the inducing points $\mathbf{z}_i \in \mathcal{S} \times \mathcal{A}$ does not take this information into account. Using a kernel for categorical features introduces issues since the actions may not be exact integers anymore. For this reason, we used the approach by [9, 14], and simply used one kernel for the states and actions.

Using a constant mean function in DGPs makes each GP mapping highly non-injective, leading to issues with the DGP prior [30]. Following [15], we used a linear mean function, $m(\mathbf{X}) = \mathbf{X}\mathbf{W}$, for all hidden layers. If input and output dimensions match, $\mathbf{W} = \mathbf{I}$; otherwise, $\mathbf{W}$ is set to the top $D^l$ left eigenvectors from the data's singular value decomposition.

### D.1.2   Neural Network Models

The linear model was treated as a single-layer feedforward network with an identity activation function:

$$\mathbf{y} = \mathbf{W}s + \mathbf{b}, \tag{26}$$

with $\mathbf{W} \in \mathbb{R}^{n \times m}$ and $\mathbf{b} \in \mathbb{R}^m$.

Regarding the choice of function approximator for the DQN algorithm, we used the following MLP as shown in Table D.1.

Note the difference in how the $q$-function is modelled compared to a GP. With a neural network, we have as input the state and an output neuron for each action, containing $\hat{q}(s,a)$. This is different for a GP, where the input is a state-action pair $z \in \mathcal{S} \times \mathcal{A}$, and the output is a Gaussian with a mean and variance for the action-value.

## D.2   Hyperparameter Tuning

Hyperparameter selection for DQN was based on pretuned settings from the `Stable-Baselines3 Zoo` GitHub repository [31]. Since the $\epsilon$-greedy schedule in `Stable-Baselines3` is based on timesteps, we adjusted to compare policies over the same number of episodes by running DQN for the recommended timesteps, recording the episodes passed, and then running the remaining episodes without training.

For GP-Q/GP-SARSA, due to the experimental nature of the algorithm and the computational complexity of certain hyperparameter settings, informal testing was performed on both CartPole and Lunar Lander to identify effective hyperparameter settings. This involved examining the reward curve on runs with different configurations. We tuned GP fitting settings, such as the initial learning rate of the Adam optimizer and how optimization over the dataset was performed at each GP update.

The hyperparameters used in the experimental setup are summarized in Tables D.3 and D.2. A moderate dataset size of 10,000 or 20,000, with each GP update involving a random subset of approximately 3,500 samples, works well when optimized in mini-batches. Using a small learning rate (0.001 or 0.005) for the Adam optimizer ensures that hyperparameters are not too tightly fitted on early inaccurate $q$-value estimates. However, the learning rate should not be too low to allow the agent to learn a meaningful policy. The discount factor $\gamma$ was set to 0.99, the same as with the DQN algorithm. The selected behavioural policy chosen was Thompson sampling, but a small comparison was made to $\epsilon$-greedy and UCB in Lunar Lander.

What was tested more rigorously was the performance difference going from an SVGP to a DGP. According to [15], a DGP with a relatively small ($\sim 100$) number of inducing points generally outperforms a single-layer DGP/SVGP with a larger number of inducing points ($\sim 500$) on regression and classification.

Increasing the number of inducing points and units improves performance, but there is a point of diminishing returns. We adopted a similar approach to [15], validating performance using an SVGP with 512 inducing points and a four-layer DGP with 128 inducing points per unit. For the number of units per layer in the DGP, we use $\min(30, D^0)$ for all inner layers, where $D^0$ is the input dimensionality, and the same RBF kernel is used for all layers.

Learning was stopped for GP-Q/GP-SARSA if a close-to-optimal policy was found, defined as achieving a return of 500 for 5 consecutive episodes in CartPole and a return of $\geq 200$ for 5 consecutive episodes in Lunar Lander.

**Table D.1.** Summary of neural network parameters. $n$ is input state dimensionality and $m$ is number of actions.

| Layer (type) | Output Shape | Param # |
|---|---|---|
| Linear (layer1) + ReLU | [batch_size, 256] | $n \times 256$ |
| Linear (layer2) + ReLU | [batch_size, 256] | $256 \times 256$ |
| Linear (layer3) | [batch_size, $m$] | $256 \times m$ |
| **Total Parameters** | | $n \times 256 + 65,536 + 256 \times m$ |

**Table D.2.** Hyperparameters for GP Models in CartPole and Lunar Lander Environments.

| Hyperparameter | CartPole | Lunar Lander |
|---|---|---|
| **Fitting (Adam)** | | |
| GP Fit Num Epochs | 1 | 1 |
| GP Fit Batch Size | 128 | 512 |
| GP Fit Num Batches | 30 | 7 |
| GP Fit Learning Rate | 0.001 | 0.005 |
| GP Fit Random Batching | True | True |
| **Exploration** | | |
| UCB Beta | NA | 1.5 |
| GP E-Greedy Steps | NA | 100,000 |
| **Model** | | |
| Discount Factor ($\gamma$) | 0.99 | 0.99 |
| Batch Size | 32 | 128 |
| Max Dataset Size/Budget | 10,000 | 20,000 |
| Kernel Type | RBF | RBF |
| Behavioral Policy | Thompson sampling | Thompson sampling |

**Table D.3.** Hyperparameters for DQN Algorithm in CartPole and Lunar Lander Environments.

| Hyperparameter | CartPole | Lunar Lander |
|---|---|---|
| Learning Rate | 2.3e-3 | 6.3e-4 |
| Batch Size | 64 | 128 |
| Buffer Size | 100000 | 50000 |
| Learning Starts | 1000 | 0 |
| Gamma | 0.99 | 0.99 |
| Target Update Interval | 10 | 250 |
| Train Frequency | 256 | 4 |
| Gradient Steps | 128 | -1 |
| Exploration Fraction | 0.16 | 0.12 |
| Exploration Final Epsilon | 0.04 | 0.1 |
| Number of timesteps | 5e4 | 1e5 |

