# OpenReview forum: "Interpretable Function Approximation with Gaussian Processes in Value-Based Model-Free Reinforcement Learning"
_NLDL.org/2025/Conference — NLDL 2025 Poster_

### Official Review · Reviewer_PyJD · 2024-10-03
**Contribution is weak and writing should be formal**

**Confidence:** 3

**Summary:**

Gaussian processes are a non-standard function approximator for reinforcement learning value functions. RL value functions are usually approximated by deep neural networks or kernel regression. The GP is a flexible non-parametric estimator and has been studied in the past in the context of RL function approximation. This paper uses SVGP and Deep GPs for off-policy and on-policy learning. These provide the advantage of uncertainty estimation and interpretability. Numerical comparisons are performed in the CartPole and Lunar Lander environments.

**Strengths:**

The core idea of the paper is interesting and deserves further studies. The experiments are done thoroughly and rigorously. The background material is thoroughly explained giving sufficient context for the reader.

**Weaknesses:**

There is a definite improvement needed in the contributions and writing before being published. While the experimental results are unfavorable, a clear illustration of the advantages offered by the paper along with more formal explanations of the math and algorithms would have pushed the paper towards a higher score. The idea of using Gaussian processes for function approximation is not new. More strong experimental evidence on other benchmarks could be useful to ascertain further.

**Final Rebuttal Confidence:**

3

**Final Rebuttal Justification:**

Based on the discussion in the committee chat and the discussion with the Area chairs, I am increasing my scores. The paper does not make a significant contribution and the results are weak. However, there is nothing wrong with the method or the evaluation. Hence, I am increasing the scores.

**Justification:**

The contribution is weak and the experimental results are not great. Formal approach to writing would have merited a higher score.

---

### Official Review · Reviewer_WsxQ · 2024-10-09
**A well-writen paper on the use of Gaussian processes as function approximators in RL with no significant shortcomings**

**Confidence:** 3

**Summary:**

In this paper, the authors introduce a novel framework based on Gaussian processes (GP) used for off-policy and on-policy learning. The main motivation is the fact that GPS offers uncertainty estimates, offering the best of both worlds - interpretability similar to that of linear models, and the capacity to capture complex patterns, like deep neural networks. This type of balance makes GPs an appealing middle ground in cases where both transparency and the ability to learn intricate relationships are pivotal. More precisely, two novel GP-based reinforcement learning algorithms are proposed, and shown to outperform linear models, and somewhat underperform DNNs. The results are sound, the paper seems technically correct, while the results themselves offer only an incremental contribution, primarily extending existing work.

**Strengths:**

- The paper appears to be technically sound with the methodology being clearly laid out
- It is also logically structured and no major errors or inconsistencies have been discovered (besides what is written under Weaknesses)
- The theoretical background (Gaussian processes, value-based RL) is well presented and understandable
- As for applicability, the authors provide comprehensive implementation details - the values of the hyperparameters are provided as well as the pseudocode with all the steps of the algorithm
- The paper contains information-rich appendices with additional results and theoretical framing, which is commendable. The additional results are also sound and complement the main part of the paper nicely.
- It is interesting (and desirable) to also see energy usage as an evaluation metric (together with space complexity)
- The paper is clearly presented and acknowledges its limitations (e.g., concerning the stability of DQN with an MLP in CartPole)
- In overall, the paper makes a decent contribution, is systematically organized, and contains no major flaws.

**Weaknesses:**

- The scientific contributions are relatively incremental (albeit still reasonable)
- The results subsection is a bit short and making it more thorough would improve the paper quality even more, with more focus on interpretability
- The literature review part is somewhat scarce - the authors could have included additional literature related to the use of GP in RL, such as "Grande R, Walsh T, How J. Sample efficient reinforcement learning with Gaussian processes. In International Conference on Machine Learning 2014 Jun 18 (pp. 1332-1340). PMLR." Despite the strict page limit, that paper should be contexualized a bit more.
- Some phrasings are awkward - parts of the paper should be rewritten. For example: "This work focuses on the advantages and limitations of GPs as function approximations for model-free RL. With a particular focus..." Clearly, this should all be part of a single sentence, or alternatively, rephrased.
- The proposed approach would benefit from being tested on a wider range of RL control environments (besides CartPole and Lunar Lander). This would provide a more robust validation of the proposed methods.

**Justification:**

This is a well-written methodologically-sound paper on the use of GPs in the context of value-based model-free reinforcement learning. The methodology is expounded on in great detail, all the hyperparameters and settings needed for implementation seem to be there, and the results show decent (expected) performance. While it doesn't significantly move the boundaries of its subfield, a while the proposed approaches should be tested on a wider range of RL environments, it is in my view a decent manuscript that is clearly above the acceptance threshold.

---

> ### Author Rebuttal · Authors · 2024-10-21
>
> Dear Reviewer, we appreciate your comments and to address them we implemented the following changes in the paper:
>
> * We added a paragraph in the introduction, incorporating reference material on inducing-point methods in Gaussian Processes (GPs) to provide context for the GP-Q algorithm and recent techniques that address the limitations of standard GPs, particularly the curse of dimensionality. The suggested reference was also included in the discussion of state-of-the-art GPs in reinforcement learning (RL). The main distinction between Grande and Walsh’s work and ours lies in the use of inducing-point methods, which are more scalable compared to the Delayed DGPQ approximation techniques used for model-free RL in the referenced work. For comparison, Delayed DGPQ tackled the control the aircraft task in $\mathcal{S} \in \mathbb{R}^{5}$ and $\mathcal{A} \in \mathbb{Z}^{3}$, whereas DeepGP and SVGP were tested on the Lunar Lander environment with $\mathcal{S} \in \mathbb{R}^{8}$ and $\mathcal{A} \in \mathbb{Z}^{4}$.
>
> * Added Sections 2.3.1, “Interpreting the GP Posterior Predictive Distribution for Action-Value Functions,” and 4.1 “Interpreting GP predictions in Lunar Lander” to explain the interpretability and UQ of GPs for action-value function estimation in RL.
>
> * Added Appendix A, “Future Work,” to discuss the next steps of the project in both the short and long term.
>
> Furthermore, although we acknowledge that the selected environments consist of simple control tasks, they serve as reliable initial benchmarks for evaluating the performance of GPs with respect to linear regression and MLP, given the constraints of the GP algorithms, i.e., discrete action spaces and continuous state spaces. As expected, GPs struggled to scale in the Lunar Lander due to the large amount of data involved in the task. Given this, exploring more complex environments with the current algorithms may not be the best approach right now. A better focus would be on modifying the algorithms to scale better in higher-dimensional environments, as outlined in the suggestions in Appendix A “Future Work.”

---

### Official Review · Reviewer_GeCz · 2024-10-11

**Confidence:** 4

**Summary:**

This paper presents a method for using Gaussian processes for approximate Q-learning. It extends GP-Q and GP-SARSA with SVGP and DGP for state-action value approximation. The experiments, though limited to two fairly simple discrete control tasks (CartPole and Lunar Lander), show that the proposed method performs better than DQNs with linear function approximations. However, DQNs with MLPs still demonstrate better performance and stability than the proposed method. The authors argue that their approach, despite lower performance compared to some baselines, is interpretable, which is an added advantage.

**Strengths:**

I view this work as a paper that combines some existing work in a fairly straightforward manner to solve the problem of action-value function approximation with interpretability on the table. While the methodological novelty may be somewhat limited, I acknowledge and appreciate the authors' efforts in synthesizing these components into a cohesive approach. I do see the value of interpretable value function approximations, and this work takes a step towards that, which I appreciate. The paper is well-written and easy to comprehend.

**Weaknesses:**

In terms of where the paper can be improved, I have a few concerns, primarily regarding the experiments and whether some of the motivations stated by the authors are sufficiently justified through the presented results. The experiments are currently limited to relatively simple control tasks, such as CartPole and Lunar Lander. I believe the authors should consider tasks with larger state and action spaces. My concern is that the performance of the proposed method, which uses GPs for value function approximation, may degrade in higher-dimensional problems, given the known limitations of GPs in handling high-dimensional problems. The current experiments do not adequately address whether this is the case and, if it is, to what extent. If this is the case, while it might be a major limitation of the method, I can still see value in the proposed method. However, in that case, I would like to see revised contributions accordingly.

Second, the authors motivate the work for why GPs are good for the value function approximation with the interpretability they offer. It is not clear to me how I should interpret GPs in value prediction. I did not find any experiments showing how this is done. Since the choice of GPs is central to the method, and given that they underperform compared to DQNs with MLPs, there needs to be a strong justification for selecting GPs as function approximators. Specifically, the interpretability offered by GPs should be clearly demonstrated to support this choice.  Perhaps I might be missing something, as the authors may have thought about this more. I would appreciate any thoughts the authors have on this or experiments showing how the interpretability is achieved and how it can be used.

As for baselines, the authors use DQN with linear functions. While this is a necessary baseline, it is also a weak baseline. Another stronger baseline could be linear function approximation with features that are non-linearly projected (ex: polynomial features, simple MLPs as encoders). Such an approach would trade off the interpretability to some extent but offer better performance.

**Justification:**

Overall, I think this work takes on an important problem, but in its current form, it falls short of novelty in the proposed method and empirical rigor in experiments, specifically in terms of number of different tasks considered and their difficulty, how well the experiments show the use of the proposed method for interpretability - the motivation of the work. A better selection of baselines could also strengthen the work. Due to these reasons, I am leaning toward rejecting the paper.

---

> ### Author Rebuttal · Authors · 2024-10-21
>
> Dear Reviewer, we appreciate your comments and would like to address the main concerns from the paper:
>
> * The proposed work aims to explore the capabilities of Gaussian Processes (GPs) as function approximators in Reinforcement Learning (RL). While GPs are well-known in supervised learning for their function approximation capabilities, with the added advantage of uncertainty quantification (UQ) derived from the variance of the Gaussian distribution. In RL, GPs require extra steps to work online, such as: **Lines 7 to 10 in Algorithm 1 and Lines 15 to 20 in Algorithm 2**: GPs require a constantly updated dataset of samples with appropriate input/output labeling to allow the model to learn the mapping from actions and states to action-value function realizations. **Line 10 in Algorithm 1 and Lines 22 to 25 in Algorithm 2**: To ensure model stability, GPs should not be updated after every episode; moreover, they require a non-trivial loss function to learn the input/output mapping.
>
> * To our knowledge, there is no detailed comparison between different GP variants, such as Deep GP and Sparse Variational GP,  and Deep RL in the literature. Our objective is not to surpass Deep RL models using multilayer perceptrons (MLPs), but to evaluate the representational and interpretability capabilities of GPs compared to other function approximators.
>
> * Although we acknowledge that the selected environments consist of simple control tasks, they serve as reliable initial benchmarks for evaluating the performance of GPs with respect to linear regression and multilayer perceptrons (MLPs), given the constraints of the GP algorithms, i.e., discrete action spaces and continuous state spaces. As expected and highlighted in your comment, GPs struggled to scale in the Lunar Lander due to the large amount of data involved in the task. Given this, exploring more complex environments with the current algorithms may not be the best approach right now. A better focus would be on modifying the algorithms to scale better in higher-dimensional environments, as outlined in the suggestions in Appendix A “Future Work.”
>
> * The experiments clearly showed that GP variants outperformed linear approximators and performed below MLPs. However, GPs provide the additional benefit of UQ in estimating the value-action function, offering a tool for interpreting the agent’s actions that MLP does not provide. As suggested in the (new) section 2.3.1, this UQ could be leveraged in safe RL, letting the agent choose not only the best action according to the value function but also the most certain one, even during training.
>
> Main changes in the paper:
>
> * Added a paragraph in the introduction with reference material on inducing point methods in GPs.
>
> * Added Sections 2.3.1, “Interpreting the GP Posterior Predictive Distribution for Action-Value Functions,” and 4.1 “Interpreting GP predictions in Lunar Lander” to explain the interpretability and UQ of GPs for action-value function estimation in RL.
>
> * Added Appendix A, “Future Work,” to discuss the next steps of the project in both the short and long term.

---

### Meta-Review · Area_Chair_Ae89 · 2024-10-31

**Recommendation:** Accept (Poster)
**Confidence:** 4

**Metareview:**

The paper presents a study that explores the use of GPs for Q-function approximation, which trades-off interpretability vs. performance. While the reviewers raised concerns about the novelty and significance of the outcomes, the analysis is sound, and the insights are useful for the RL community. Hence, with the expectation that the camera-ready version will place a stronger focus on interpretability to better substantiate the claims, I recommend accepting the paper.

**Suggested Changes To The Recommendation:**

1: I agree that the recommendation could be moved down

---

### Decision · Program_Chairs · 2024-11-06

**Decision:**

Accept (Poster)

**Comment:**

We recommend a poster presentation given the AC and reviewers recommendations.